# Toxin Homology Domain in Plant Type 2 Prolyl 4-Hydroxylases Acts as a Golgi Localization Domain

**DOI:** 10.3390/cells13141170

**Published:** 2024-07-09

**Authors:** Ryo Moriguchi, Ken Matsuoka

**Affiliations:** Department of Bioscience and Biotechnology, Faculty of Agriculture, Kyushu University, Fukuoka 819-0395, Japan

**Keywords:** hydroxyproline, cysteine-rich domain, protein targeting, lumenal protein, Golgi

## Abstract

Prolyl 4-hydroxylase (P4H) generates hydroxyproline residues in proteins. Two classes of P4H have been found in plants. Type 1 P4H has a signal anchor at the N-terminus, while type 2 P4H has both an N-terminal signal peptide and a C-terminal toxin homology domain (Tox1 domain) with six conserved cysteine residues. We analyzed the localization of tobacco type 2 P4H (NtP4H2.2) in tobacco BY-2 cells. Cell fractionation studies, immunostaining of cells, and GFP fusion study indicated that NtP4H2.2 localizes predominantly to the Golgi apparatus and is a peripheral membrane protein associated with the luminal side of organelles. Expression of the GFP-Tox1 domains of NtP4H2.2 and another tobacco type 2 P4H NtP4H2.1 in BY-2 cells and Arabidopsis epidermal cells indicated that these proteins were targeted to the Golgi. The Tox1 domains from Arabidopsis and rice type 2 P4Hs also directed GFP to the Golgi in tobacco BY-2 cells. The Tox1 domain of NtP4H2.2 increased the membrane association of GFP, and mutation of the cysteine residues in this domain abolished Golgi localization. Furthermore, the catalytic domain of NtP4H2.2 also directed GFP to the Golgi. Thus, the Tox1 domains of plant P4Hs are the Golgi localization domains, and tobacco P4H2.2 localizes to the Golgi by the action of both this domain and the catalytic domain.

## 1. Introduction

Many proteins transported through the secretory pathway are subjected to post-translational modifications. Peptidyl proline oxygenation that generates 4-hydroxyproline residues is historically called proline hydroxylation and occurs in both plants and animals. In plants, the consensus sequence for such modification is complex [1], and such hydroxyproline residues in cell wall proteins act as target sites for O-glycosylation [2]. One such family of O-glycosylated proteins in plants is that of the arabinogalactan proteins. Arabinogalactan proteins are hydroxyproline-rich plant proteoglycans and contain large, branched arabinogalactan polysaccharide chains attached to their hydroxyproline residues [3]. Another group of cell wall proteins rich in glycosylated hydroxyproline is extensin. In this case, hydroxyproline residues are glycosylated with mono- and oligo-arabinose [3]. Likewise, vacuolar proteins and some secretory peptide hormones contain hydroxyproline residues, and some of them are glycosylated with either arabinogalactan-type glycan or oligo-arabinose [2,4].

The enzyme that catalyzes the conversion of proline to hydroxyproline residues in protein is called prolyl-4 hydroxylase (P4H). P4H activities are necessary for the triple helix formation of collagen in the ER of vertebrate cells. In animals, this enzyme is found in the lumen of the ER in a soluble form. This enzyme consists of a catalytic α subunit and a non-catalytic β subunit, which is identical to protein disulfide isomerase (PDI). The function of PDI in the P4H complex includes keeping the α- subunits in a soluble form and catalytically active [5] and retaining them in the ER as mediated by a C-terminal K/HDEL ER retention signal. On the other hand, previous studies indicated that plant P4Hs do not form a complex with PDI [6,7,8,9]. As O-glycosylation to hydroxyproline in plant cells starts at the Golgi apparatus or the transitional organelle between the ER and the Golgi [10], proline residues in proteins should be converted before O-glycosylation starts. Therefore, an alternative mechanism that is independent of KDEL and PDI might exist to retain plant P4Hs in the ER or Golgi.

Plants have multiple P4Hs. For example, *Arabidopsis thaliana* has 11 P4H homologs [8]. Plant P4Hs are divided into two groups based on their primary structures. Type 1 P4Hs are type II membrane proteins that have an N-terminal signal anchor followed by a catalytic domain. Our characterization of NtP4H1.1 (formally called tobacco PH) [8], which is a type 1 P4H of tobacco, revealed that it is located in the Golgi apparatus [8]. The other classes of plant P4Hs, which are called type 2 P4Hs, have an N-terminal short hydrophobic stretch followed by a catalytic domain and a C-terminal toxin homology (Tox1) domain of approximately 40 amino acids [7]. The Tox1 domain has homology with the ShK toxin from *Stichodactyla helianthus* [11] and the BgK toxin from *Bunodosoma granulifera* [12,13], with six conserved cysteine residues, which are connected by disulfide bonds [12]. Three type 2 P4Hs, including At-P4H-2 [7], Cr-P4H-1 [9], and DcP4H1 and 2 [14], were examined for enzymatic activity.

Previously, we have shown that the N-terminal cytosolic domain, as well as the protein–protein interactions between catalytic domains, define the intracellular localization of type 1 P4Hs [8,15]. We also showed that a fusion protein of GFP and an Arabidopsis type 2 P4H AtP4H2 is targeted to the Golgi [15]. However, the mechanism of the intracellular localization of type 2 P4Hs and the function of the Tox1 domain has not been elucidated yet. In this study, we report the intracellular localization of tobacco type 2 P4Hs and characterize the role of Tox1 domains and catalytic domains on the localization.

## 2. Materials and Methods

### 2.1. Cell Culture

Cultures of tobacco BY-2 cells were maintained as described [16]. The cell lines we used were wild-type BY-2, BY-2 cells that express NtSCAMP2-mRFP, NtP4H1.1-mRFP, or NtP4H1.1-mRFP and NtSCAMP2-YFP [17]. For fluorescent microscopy, log phase cells (3 or 4 days after subculture) were used, whereas in the protein extraction and microsomal preparations, we used cells 5 days after subculture.

### 2.2. cDNA Cloning of Type 2 P4Hs, Construction of Plasmids, and Transformation of Tobacco BY-2 Cells and Arabidopsis

Full-length cDNAs for NtP4H2.1 (accession No. AB471926) and NtP4H2.2 (accession No. AB471927) were isolated from a tobacco full-length enriched cDNA library [17] using EST information from DV999364 and EB427781, respectively, as described previously [17]. The amino acid sequences were aligned using the CLUSTALW ver. 2.1 program “https://www.genome.jp/tools-bin/clustalw (accessed on 7 September 2006)”. To create the fusion proteins with GFP, a secretory derivative of GFP (spo41(I28G, P36Q)-GFP) [1] was used. The fusion of either the coding regions for the signal-peptide-omitted type 2 P4H or for the Tox1 domain was amplified by PCR and cloned into the KpnI-EcoRI sites of pMAT328, which contains KpnI and EcoRI sites following the coding region of the secretory derivative GFP, driven by the CaMV35S promoter. Transformations with these constructs of tobacco BY-2 cells were performed as described [16]. In some cases, transiently expressed protein was analyzed as follows: Two to three days after the start of the co-culture of Agrobacterium and tobacco BY-2 cells, tobacco cells were washed with a liquid medium and incubated in a liquid medium containing 500 mg/L cefotaxime. After two days, the fluorescence of the expressed protein was analyzed under a fluorescence microscope. The transient expression of these constructs and NtP4H1.1-mRFP in Arabidopsis leaf epidermal cells were performed using the method described by [18] with a minor modification; the condition of vacuum infiltration of Agrobacterium solutions into seedlings was changed to 3 or 4 times with the vacuum condition of −0.06 kPa for 1 min each.

The Tox1 domain of rice and Arabidopsis were also cloned into the KpnI-EcoRI site of pMAT328, as described above. In these cases, the PCR fragments corresponding to the Tox1 domains of AK059759, AK103739, At-P4H-2, and AT3G28480 were amplified by PCR using rice or Arabidopsis cDNAs as templates. These constructs were expressed in tobacco BY-2 cells, which were already expressing NtP4H1.1-mRFP, as described above.

### 2.3. Antibodies

Polyclonal anti-NtP4H2.2 antibody was raised in rabbits against the mixture of synthetic peptides (CSGWHNDKKTKSSVLKL and NSEAKKTQAKGDDWSDC) conjugated with KLA and used after affinity purification using immobilized antigen peptides. Polyclonal anti-GAPDH antibody was raised in rabbits against a KLH-conjugated mixture of peptides, CTQKTVDGPSMKDWRGGR and CKEESEGKLKGILGFTED, which correspond to conserved regions of GAPDHs from >50 higher plant species. The other antibodies are as described [8].

### 2.4. Quantification of GFP Fluorescence in Total Culture, Cell, and Medium Fractions

The fluorescence intensity of the whole culture was measured as follows: Cell culture was mixed with concentrated PBS, water, and Triton X-100 to give the final concentration of PBS pH7.4 and 1% Triton X-100. The suspension was sonicated as described [16]. To monitor the fluorescence intensity of cells, cells were separated from the medium by filtration, and cells were mixed with buffer containing Triton X-100 to give a final concentration of PBS pH7.4 and 1% Triton X-100, and sonicated as above. The recovered medium was mixed with buffers as above and used for the quantification. The fluorescence of GFP was measured using a Spectramax Gemini fluorometer (Molecular Devices, San Jose, CA, USA) with 488 nm excitation and 520 nm emission. The significance of the difference in fluorescence intensities was analyzed by ANOVA.

### 2.5. Preparations and Separations of Microsomes

All steps were performed at 4 °C unless otherwise stated. Microsomes were prepared from BY-2 cells as follows: BY-2 cells were homogenized in equal amounts of extraction buffer consisting of 40 mM Hepes-KOH (pH 7.4), 0.4 M sorbitol, 0.1 M KOAc, 6 mM EGTA, 4 mM EDTA, 20% glycerol, 2 mM DTT, and one tablet of complete mini-protease inhibitor cocktail EDTA free (Roche, Mannheim, Germany). The homogenate was centrifuged at 1000× *g* for 10 min to yield a total protein fraction, followed by ultracentrifugation at 100,000× *g* for 60 min to yield a soluble protein fraction. The precipitate of this centrifugation was resuspended in an equal amount of resuspending buffer consisting of 50 mM HEPES-KOH (pH 6.8), 20% glycerol, and 1 mM DTT, and used as an insoluble fraction.

For the preparation of microsomes, the total protein fraction was centrifuged at 10,000× *g* for 10 min, followed by subsequent ultracentrifugation at 100,000× *g* for 60 min. The precipitate was resuspended in the resuspending buffer. The protease digestion assay was performed as described in [8]. The microsomal sonication assay was performed as follows: Microsome fractions were mixed with buffers and then incubated on ice for 30 min. Next, the mixture was subjected to two 30 s sonication pulses at level 2.5 with a Bioruptor UCD-200TM sonicator (COSMO BIO, Tokyo, Japan). Soluble and membrane fractions were separated by ultracentrifugation (100,000× *g*, 60 min).

Organelles in microsomal fractions were separated by isopycnic ultracentrifugation, as described in [19].

Immunoblot detections of protein were performed using the following antibodies: Sec61 (1:1000), Sec22 (1:1000), NtP4H2.2 (1:1000), GAPDH (1:1000), and PDI (1:1000). The antigen–antibody complex was detected using ECL Plex goat-α-rabbit IgG-Cy5 (GE healthcare) at a dilution of 1:2500 and the signals on membranes were detected by a Typhoon 9400 image analyzer (Molecular Devices).

The fluorescence of GFP or RFP fusion proteins, separated after SDS-PAGE, was detected, as described in [17], using a Typhoon 9400 image analyzer.

### 2.6. Preparation of Protoplasts, Immunoblotting, and Peroxidase Assay

Protoplasts were prepared from 3-day-old BY-2 cells, as described in [20], with several modifications. Briefly, protoplasts obtained after the enzymatic degradation were washed with BY-2 medium, including 0.4 M mannitol and 10 mg/L 2,6-dichlorobenzonitrile. After washing, the cell density was adjusted to 1.0 × 10^6^ cells/mL with the same solution, and 4 mL of these cells were placed into a 9 cm diameter dish. After 3 days of incubation at 28 °C, their cell and medium fractions were collected. Total proteins of the cell fraction were extracted as described below: cells were resuspended in 1xPBS equivalent to the volume of the medium fraction and sonicated for 2.5 min. After centrifugation at 1000× *g* for 10 min at 4 °C, the supernatant was collected as total cellular proteins.

Protein distributions in each fraction were determined by immunoblot analysis with the antibodies described above. The peroxidase assay was performed using a TMB Microwell Peroxidase Substrate System Kit (Kirkegaard and Perry Laboratories, Gaithersburg, MD, USA) according to the manufacturer’s instructions using horseradish peroxidase (Wako, Osaka, Japan) as a standard.

### 2.7. Fluorescent Microscopy

Tobacco cells were mounted on a slide, and then their GFP or RFP fluorescence was observed by an IX81 fluorescent microscope (Olympus, Tokyo, Japan) equipped with an IX2-DSU module (Olympus) that generates confocal images. Images were captured by an iXon DU-888E CCD camera (Andor, Belfast, Northern Irland) using either GFP or RFP filter and mirror sets and then processed using the MetaMorph version 5.0.3 software package (Molecular Device). For colocalization frequencies of GFP and RFP signals, GFP and RFP fluorescence intensities in fluorescent dots were obtained by the Linescan program in the MetaMorph software. The percentage of the area of the green signal that overlapped with the red signal was used for the colocalization frequency. The average of the colocalization frequencies of GFP and RFP signals in randomly chosen dots from 18 independent transformants was calculated. For the detection and comparison of transiently expressed fluorescence of different constructs in tobacco BY-2 cells, cells showing roughly similar brightness were chosen to record images.

For BFA treatment, a 5 mg/mL stock solution of BFA in DMSO was prepared. BY-2 cells were treated with 5 mg/mL (14.8 µM) BFA for 120 min at room temperature with gentle shaking. Control treatments were performed with equal amounts of DMSO.

For immunostaining, cells were fixed with formaldehyde and stained with primary and secondary antibodies as described in the supplemental methods of Ref. [21]. Stained cells were incubated overnight in a 50% glycerol solution at 4 °C and then mounted on slides. Images were collected using a confocal laser scanning microscope (TCS SP8 with Lightning, Leica, Wetzlar, Germany) installed at the Center for Advanced Instrumental and Educational Supports, Faculty of Agriculture, Kyushu University, basically following the method described in Ref. [21].

## 3. Results

### 3.1. Cloning of Type 2 P4H cDNAs from Tobacco Cells and Comparison of the Primary Structure against Plant P4Hs

#### 3.1.1. Cloning of Type 2 P4H cDNAs from Tobacco Cells

To clone the cDNAs of tobacco type 2 P4Hs, we searched an EST database in the public domain “http://www.ncbi.nlm.nih.gov/dbEST/index.html (accessed on 18 May 2006)” and found four tobacco ESTs (accession nos. DV999364, EB427781, EB437175, and EB447902), which encode proteins that showed high sequence similarity to type 2 P4H homologs of Arabidopsis and rice. Then, we cloned the full-length cDNAs corresponding to DV999364 and EB427781 and obtained cDNAs for NtP4H2.1 and NtP4H2.2, respectively, from a complete-length enriched cDNA library of tobacco BY-2 cells [17].

#### 3.1.2. Comparison of Amino Acid Sequences

Sequence analyses and deduced amino acid sequences of these cDNAs revealed that both contained an N-terminal hydrophobic amino acid stretch, a catalytic domain for prolyl hydroxylase, which contains a conserved HxD motif and histidine, and a C-terminal Tox1 domain [Appendix A]. Their deduced amino acid sequences revealed that NtP4H2.1 and NtP4H2.2 contain 294 and 318 amino acid residues, respectively, and have 61% identity with each other. They have a relatively high similarity with type 2 P4Hs in other plants, but their similarity with type 1 P4Hs is limited.

The N-terminal hydrophobic stretches were shorter in type 2 P4Hs, including NtP4H2.1 and NtP4H2.2, than in type 1 P4Hs. The hydrophobic stretches of tobacco type 2 P4Hs were predicted as cleavable signal peptides by SignalP 4.0 “http://www.cbs.dtu.dk/services/SignalP/ (accessed on 18 August 2012) [22], whereas the longer hydrophobic stretch in NtP4H1.1 was predicted as transmembrane region and is actually a type II signal anchor [8]. Although the Tox1 domains from NtP4H2.1 and NtP4H2.2 contain six conserved cysteine residues similar to the BgK and ShK toxins, the sequence similarity between these Tox1 domains and the toxins is low (24% and 22%, respectively) (Appendix A).

### 3.2. Membrane Topology and Intracellular Localization of Tobacco Type 2 P4Hs

#### 3.2.1. Organelle Association and Membrane Topology of Endogenous NtP4H2.2

To analyze the membrane topology and intracellular localization of tobacco type 2 P4Hs, we first prepared a specific antibody against NtP4H2.2 using oligopeptides that are unique to this protein. To determine the specificity of the antibody, total proteins extracted from tobacco BY-2 cells were separated by SDS-PAGE and subjected to immunoblot analysis using the anti-NtP4H2.2 antibody. A band of 33.9 kDa corresponding to the predicted molecular mass of NtP4H2.2 without a signal peptide was observed (Appendix A). No such band was detected when the same proteins were probed with pre-immune serum (Appendix A). Thus, we concluded that the antibody specifically recognizes NtP4H2.2. We also tried to make a specific antibody against NtP4H2.1 using the same method but were not successful. Therefore, we focused on characterizing NtP4H2.2 by immunological methods.

We first analyzed whether some of the NtP4H2.2 was secreted to the medium because the precursor to type 2 P4H contains a typical signal peptide. We prepared medium and cellular protein fractions from tobacco BY-2 cell cultures and subjected them to immunoblot analysis. We observed a 33.9 kDa band in the cell fraction. No such signal was observed in the culture medium (Figure 1A). To exclude the possibility that NtP4H2.2 was trapped in the cell wall, we prepared protoplasts from BY-2 cells and determined the distribution of NtP4H2.2 after culturing the protoplasts in a medium containing an inhibitor for cell wall biosynthesis. NtP4H2.2 was recovered almost completely in the cell fraction (Figure 1B). Under the same conditions, almost all the protein disulfide isomerase (PDI: soluble protein in the ER lumen) and glyceraldehyde 3-phosphate dehydrogenase (GAPDH: soluble protein in the cytoplasm) were recovered in the cell fraction, whereas about half of the enzymatic activity of peroxidase, which is a secreted protein found in cell walls, was detected in the medium fraction (Figure 1B). This observation confirmed that NtP4H2.2 is localized in the cell.

We next examined whether NtP4H2.2 was associated with membranes. Total cell lysates from BY-2 cells were separated by centrifugation, and the distribution of NtP4H2.2 in soluble and precipitated fractions was examined by immunoblot analysis. NtP4H2.2 was recovered in the precipitated fraction (Figure 1C). Next, we determined the topological orientation of NtP4H2.2 in the membrane using a protease protection assay. Microsomes from transformed BY-2 cells expressing NtP4H1.1-mRFP (formally called PH-mRFP) [17] were treated with trypsin in the presence or absence of Triton X-100. Then, the susceptibilities to trypsin of NtP4H2.2, NtP4H1.1-mRFP, and Sec22 were examined. NtP4H1.1 is a type II membrane protein with a large luminal domain, whereas Sec22 is an R-SNARE protein that has a large N-terminal cytosolic domain. Unlike Sec22, NtP4H2.2 was only digested by trypsin in the presence of detergent, as was the case for NtP4H1.1-mRFP (Figure 1D). These results indicated that NtP4H2.2 is located at the luminal side of the membranous organelle(s).

We then analyzed the membrane association of NtP4H2.2. Microsomes were sonicated in buffers that contained either 1 M NaCl, 2.5 M urea, 0.1 M Na_2_CO_3_, or 1% Triton X-100. Then, the soluble and precipitated fractions were separated by ultracentrifugation. In the presence of a chaotropic agent (urea), Na_2_CO_3_, or a high concentration of salt, some of the NtP4H2.2 was recovered in the soluble fraction (Figure 1E). In other cases, almost all the NtP4H2.2 was recovered in the pellet. Under the same conditions, Sec61, which is an integral membrane protein in the ER, was recovered in the pellet, and most of the PDI, which is a soluble protein in the ER lumen, was recovered in the soluble fraction (Figure 1E). These data indicated that NtP4H2.2 is a peripheral membrane protein tightly associated with the membrane.

#### 3.2.2. Intracellular Localization of Endogenous NtP4H2.2

We also investigated the intracellular localization of NtP4H2.2 by subcellular fractionation. Microsomal membranes were separated by isopycnic sucrose density gradient centrifugation followed by SDS-PAGE and immunoblot analyses. To facilitate the analysis, we used transgenic BY-2 cells expressing organelle marker proteins fused with fluorescent proteins. We used NtP4H1.1-mRFP as a marker for the ER and Golgi since tobacco type 1 P4H and its fusion proteins cycle between the Golgi and the ER [8,17]. We also used NtSCAMP2-YFP [17] as a marker for the trans-Golgi network (TGN), secretory vesicle cluster (SVC), and plasma membrane. In the presence of EDTA, the distribution of NtP4H2.2 was nearly identical to that of NtP4H1.1-mRFP and was distinct from that of NtSCAMP2-YFP (Figure 2A). In the presence of MgCl_2_, the distribution of NtP4H2.2 and NtP4H1.1, as well as NtSCAMP2-YFP, shifted to higher-density fractions. The migration pattern of NtP4H2.2 was also nearly identical to that of NtP4H1.1-mRFP and was distinct from that of NtSCAMP2-YFP (Figure 2A). These data indicated that NtP4H2.2 is localized in a similar place to NtP4H1.1 [8].

To further investigate the localization of NtP4H2.2, fixed tobacco BY-2 cells were stained with anti-NtP4H2.2 antibody, and an optical image of the stained signal was collected. Optical section images of stained cells collected by the Leica lightning method showed punctate signals that are scattered throughout the cytoplasm (Figure 2B). This distribution pattern is a typical pattern of the distribution of the Golgi apparatus in tobacco BY-2 cells [8].

#### 3.2.3. Intracellular Localization of GFP-Tagged NtP4H2.1 and NtP4H2.2

We tagged NtP4H2.1 and NtP4H2.2 with a secretory derivative of GFP, and the resulting fusion proteins, GFP-NtP4H2.1 and GFP-NtP4H2.2 (Appendix A), were expressed in tobacco BY-2 cells expressing mRFP-tagged organelle markers to further confirm the Golgi localization. We used spo41(I28G, P36Q)-GFP (Appendix A) as a secretory derivative of GFP because this protein is predominantly secreted into the medium, but some transport intermediate forms can be observed in the Golgi, TGN, and SVCs [1,17]. The GFP-NtP4H2.1 or GFP-NtP4H2.2 proteins were expressed in tobacco cells expressing NtP4H1.1-mRFP. Both the GFP-NtP4H2.1 and GFP-NtP4H2.2 proteins showed punctate localization where NtP4H1.1-mRFP localized (Figure 3A,B). Although their distribution resembled that of spo41(I28G, P36Q)-GFP (Figure 3C), their colocalization frequencies with NtP4H1.1-mRFP were higher than that of spo41(I28G, P36Q)-GFP (Figure 3D). When GFP-NtP4H2.1 or GFP-NtP4H2.2 was expressed in cells expressing NtSCAMP2-mRFP (Figure 3E,F), little colocalization of the GFP and mRFP dots (3.6% and 5.2%, respectively) was observed. Interestingly, many dots of these colors were closely associated (Figure 3E,F). These results and the results of the cell fractionation and immunostaining studies (Figure 2) indicate that these tobacco type 2 P4Hs are predominantly localized in the Golgi.

### 3.3. Role of Tox1 Domain in Subcellular Distribution

#### 3.3.1. Role of Tox1 Domain in the Subcellular Distribution of NtP4H2.1 and NtP4H2.2

Although no predicted membrane-spanning region was present in the NtP4H2.2 sequence, this protein localized in the luminal side of the Golgi and tightly associated with membranes (Figure 1 and Figure 2). In addition, GFP-NtP4H2.1 or GFP-NtP4H2.2 was localized to the Golgi (Figure 3). Thus, there should be a mechanism that directs these proteins to their destination. The type 2 P4Hs have a common feature, that is, all these proteins have a Tox1 domain. Since the function of this domain is unknown, we analyzed its role in protein localization. First, we made constructs that contain spo41(I28G, P36Q)-GFP fused with the Tox1 domain from either NtP4H2.1 or NtP4H2.2, and the resulting proteins, GFP-Tox1(NtP4H2.1) and GFP-Tox1(NtP4H2.2) (Appendix A), were expressed in transformed tobacco BY-2 cells that express mRFP-tagged organelle markers. As shown in Figure 4, punctate fluorescence of both GFP-Tox1(NtP4H2.1) and GFP-Tox1(NtP4H2.2) was seen in transformed cells regardless of the comparison counterpart proteins with RFP (Figure 3A–D, right). The pattern of the distribution of these green dots was similar in all these cases. These punctate GFP signals were well colocalized with that of NtP4H1.1-mRFP (Figure 4A,B right and Appendix A). Their colocalization frequencies with NtP4H1.1-mRFP (72% and 66%, respectively) were higher than that of spo41(I28G, P36Q)-GFP (51%). We introduced the same constructs into BY-2 cells expressing NtSCAMP2-mRFP. Almost all the punctate signals did not colocalize with NtSCAMP2-mRFP dots, but some of them are localized in the close vicinity of the NtSCAMP2-mRFP puncta. (Figure 4C,D right). As some of the NtSCAMP2-mRFP dots represent the TGN [17], this localization pattern also suggests that GFP-Tox1 fusions were targeted to the Golgi.

We then compared the distribution of GFP-Tox1(NtP4H2.2) with that of endogenous NtP4H2.2 in tobacco BY-2 cells using subcellular fractionation methods. Microsomal fractions from transformed cells were separated by isopycnic sucrose density gradient centrifugation, and then the distribution of proteins in the gradient was compared. The distribution of GFP-Tox1(NtP4H2.2) was almost identical to that of NtP4H2.2. (Figure 4E). This result also indicated that the Tox1 domain has the ability to direct the protein to the organelles where NtP4H2.2 localizes, namely the Golgi.

We examined the effects of brefeldin A (BFA) on GFP-Tox1 localization in BY-2 cells. BFA inhibits protein trafficking by the inhibition of vesicle formation at the Golgi apparatus and other membranes [23]. We previously reported that treatment of BY-2 cells with 14.8 μM BFA for 120 min changes the distribution of Golgi proteins to the ER [8] and causes mis-localization of trans-Golgi/TGN proteins to the plasma membrane [17]. After BFA treatment, the distribution of GFP-NtP4H2.2 as well as GFP-Tox1(NtP4H2.2) changed from Golgi to perinuclear and reticular distribution (Figure 4F and Appendix A). This distribution is quite similar to the typical distribution of the ER in tobacco BY-2 cells (Appendix A). Likewise, the distribution of GFP-Tox1(NtP4H2.2) also changed to the ER (Figure 4G and Appendix A). Identical results were obtained with GFP-NtP4H2.1 and GFP-Tox1(NtP4H2.1) (Appendix A). Under the same conditions, the localization of spo41(I28G, P36Q)-GFP changed to the cell periphery (Appendix A). These results also support the notion that the Tox1 domain has the ability to localize proteins to the Golgi.

We also performed localization analysis of GFP-Tox1 proteins in another plant, *Arabidopsis thaliana*. Agrobacterium harboring either the GFP-Tox1(NtP4H2.1) or the GFP-Tox1(NtP4H2.2) construct was mixed with Agrobacterium harboring NtP4H1.1-mRFP construct, and the resulting bacterial suspension was vacuum infiltrated into seedlings of Arabidopsis. Thereafter, transient expressions of these proteins in leaf epidermal cells were observed using a confocal fluorescence microscope. Punctate signals of GFP-Tox1 proteins colocalized with NtP4H1.1-mRFP (Figure 5A,B). These results suggested that Tox1 domains of tobacco also directed GFP to the Golgi in Arabidopsis cells.

#### 3.3.2. Tox1 Domain Increases Membrane Association

The effect of the Tox1 domain on membrane association was investigated using GFP-Tox1(NtP4H2.2) as an example and spo41(I28G, P36Q)-GFP as a control. We first analyzed whether some of the GFP-Tox1(NtP4H2.2) is secreted into the medium. Fluorescence of GFP in the whole culture, medium, and cells was quantified using a spectrophotometer. Although the whole culture of non-transformed cells showed significant autofluorescence, expression of the GFP fusions caused higher fluorescence intensities in the whole culture (Figure 6A, left). Most of the GFP-Tox1(NtP4H2.2) fluorescence was recovered in the cell fraction, whereas about half of the fluorescence of Spo41(I28G,P36Q)-GFP was recovered in the medium (Figure 6A middle and left). This observation indicated that GFP-Tox1 (NtP4H2.2) is retained in the cells.

The total cell lysate was prepared from transformed tobacco cells expressing either of the proteins, and then soluble and insoluble fractions were separated by ultracentrifugation. The distributions of spo41(I28G, P36Q)-GFP or GFP-Tox1(NtP4H2.2) in the cells were examined by immunoblot analyses. Almost all the GFP-Tox1(NtP4H2.2) were recovered in the precipitated fraction, whereas a large amount of spo41(I28G, P36Q)-GFP was recovered in the soluble fractions (Figure 6B,C). To analyze their membrane association, microsomes from the transformed cells were subjected to a membrane sonication assay as described in the legend in Figure 1E. Nearly all the GFP-Tox1(NtP4H2.2) was recovered in the pellets under all buffer conditions tested (Figure 6D). These results indicate that the fusion of the Tox1 domain to GFP increases the association of this protein to membranes.

#### 3.3.3. Tox1 Domain from Other Plant Species

From the results obtained above, the Tox1 domain of tobacco type 2 P4Hs functions as a membrane-association domain that directs proteins to the Golgi region in at least two different plant species. Therefore, we thought that the function of the Tox1 domain would be conserved in different plant species. To identify proteins that have a Tox1 domain homologous to that of NtP4H2.2, we searched for proteins with sequence similarity to the Tox1 domains of NtP4H2.2 in the NCBI database by employing the blastp method using non-redundant protein databases. We found 3147 proteins from 580 organisms that showed an E-value less than 0.05 (Appendix A). Among them, proteins with sequences showing high similarities (E < 10^−5^) are widely distributed throughout the plant kingdom, and most of the annotated proteins among them are P4H. In these Tox1 domains, more than 93% of these sequences contain six conserved cysteine residues. Proteins encoding less similar sequences are found in a wide range of eukaryotic kingdoms, including plants and non-vertebrates.

Next, we investigated whether the function of the Tox1 domain in plants is conserved. We cloned two partial cDNAs for the Tox1 domain of type 2 P4H in rice (AK059759 and AK103739) and two from Arabidopsis (AtP4H2 and AT3G28480), made GFP fusion constructs (Appendix A) and expressed them in transformed tobacco BY-2 cells expressing NtP4H1.1-mRFP. Similar to the results described above, these fusion proteins were well colocalized with NtP4H1.1-mRFP (Figure 7A–D and Appendix A). Additionally, their colocalization frequencies with NtP4H1.1-mRFP were higher than that of spo41(I28G, P36Q)-GFP (Figure 7E). These data indicated that these Tox1 domains could direct proteins to the Golgi region and imply that Tox1 domains from plants have the ability to direct proteins to the Golgi.

### 3.4. Role of the Catalytic Domain in Golgi Localization

We also examined the role of the catalytic domain of NtP4H2.1 and NtP4H2.2 in Golgi localization, as we found that the catalytic domain of NtP4H1.1 plays a role in intracellular localization [24]. Tobacco BY-2 cells expressing NtP4H1.1-mRFP were transformed with Agrobacterium harboring either the GFP-NtP4H2.1(ΔTox1) or GFP-NtP4H2.2(ΔTox1) (Appendix A), and their subcellular localizations were determined by confocal fluorescence microscopy (Figure 8). Interestingly, their punctate GFP signals colocalized with the red fluorescent signals of NtP4H1.1-mRFP (Figure 8A and Appendix A). Likewise, GFP-NtP4H2.1(ΔTox1) colocalized with NtP4H1.1-mRFP (Appendix A). These observations also suggested that the catalytic domain of type 2 P4H has the ability to localize the protein to the Golgi. Additionally, after the BFA treatment of the cells, the signals of GFP-NtP4H2.1(ΔTox1) or GFP-NtP4H2.2(ΔTox1) changed from a Golgi-localized pattern to an ER-localized pattern, like NtP4H1.1-mRFP (Figure 8B and Appendix A). Therefore, plant type 2 prolyl hydroxylases are likely to have at least two distinct signal(s) or regions for Golgi localization: the catalytic domain and the Tox1 domain.

### 3.5. Role of Cysteines in the Tox1 Domain in Golgi Localization

As almost all the cysteine residues in the Tox1 domains are conserved (Appendix A) and in many cases, such cysteine residues in luminal and secretory proteins play a role in the structural determination of protein domains by forming disulfide bridges, we questioned whether the cysteine residues in the Tox1 domain play a role in Golgi localization. The disulfide bridges of the Tox1 domain from ShK toxin were determined [12,13]. Using this information, we predicted the disulfide bridges in the Tox1 domain of NtP4H2.1 (Figure 9 upper). We expressed three mutant GFP-Tox1(NtP4H2.1) proteins that contain cysteine-to-serine mutants at different positions of the Tox1 domain in transformed tobacco cells expressing NtP4H1.1-mRFP (Figure 9).

Perinuclear and intracellular mesh-like structures of GFP fluorescence, which indicate the localization of GFP fusion proteins in the ER, were observed in cells expressing either of these three mutants, whereas such a pattern was rarely seen in cells expressing GFP fused with wild-type Tox1 domain. In addition, a few puncta of GFP were observed in cells expressing GFP-Tox1(NtP4H2.1)C291S, indicating that this protein was predominantly localized in the ER. In contrast, GFP-Tox1(NtP4H2.1)C287S and GFP-Tox1(NtP4H2.1)C294S showed both perinuclear (ER) and punctate structures overwrapping with the Golgi-distributing NtP4H1.1-mRFP signals. These observations suggested that disulfide bridges in the Tox1 domain could be important for protein folding, which subsequently affects function.

## 4. Discussion

We found that tobacco type 2 P4Hs localize on the lumenal side of the Golgi apparatus and are tightly associated with the membranes of these organelles (Figure 1, Figure 2 and Figure 3). Interestingly, the type 2 P4H sequence does not have a transmembrane region but instead has a signal peptide that allows the protein to migrate into the endoplasmic reticulum lumen. Our analyses indicated that the Golgi localization is likely to be mediated by both the catalytic domain and the Tox1 domain in their C-terminus (Figure 4 and Figure 8). The function of the Tox1 domain is not specific for the P4H catalytic domain as secretory GFP-Tox1 fusions are targeted to the Golgi in both tobacco and Arabidopsis cells (Figure 4, Figure 5 and Figure 6). In addition, Tox1 domains from both Arabidopsis and rice also targeted secretory GFP to the Golgi in tobacco cells (Figure 7). BFA treatment directed these GFP-tagged proteins to the ER (Figure 4F,G). We also observed a similar BFA effect on tobacco type 1 P4H NtP4H1.1 [8]. Furthermore, we, as well as other groups, have observed similar BFA effects on several integral membrane proteins in tobacco BY-2 cells: these include Golgi-localizing methyltransferase [25], mannosidase1 [26], mRFP-tagged sialyltransferase [27], and a SNARE protein SYP41 [28]. However, this BFA-induced redistribution of Golgi proteins to the ER is not ubiquitous among proteins, as SYP31, a SNARE protein that recycles between the ER and the Golgi, shows a punctate distribution upon BFA treatment in BY-2 cells [28]. As the tobacco type 2 P4Hs analyzed in this work are peripheral membrane proteins, there should be a Golgi-localizing receptor protein, which is an integral membrane protein that can be redirected to the ER by BFA treatment.

The proline hydroxylation catalyzed by P4Hs should proceed with the glycosylation of hydroxyproline residue. We found previously that the enzymatic activity of peptidyl hydroxyproline galactosyltransferase, the substrate of which is the product of P4H, is localized in the ER and the Golgi in Arabidopsis cells [29]. The peptidyl hydroxyproline galactosyltransferase is also reported to be predominantly targeted to the Golgi using fluorescence-protein tagging techniques [30,31,32,33]. Therefore, an attractive hypothesis is that the type 2 P4Hs are distributed with both the ER and Golgi or are recycled between these compartments to supply substrates for these glycosyltransferases. In future analyses, it will be interesting to figure out which scenario is likely.

Our analysis of the localization of Arabidopsis P4Hs using the GFP-fusion technique also indicated that these Arabidopsis proteins are likely to be localized in the ER and the Golgi apparatus [15]. Therefore, the Tox1 domain is a localization domain of luminal proteins to the Golgi, and the function of the Tox1 domain is likely to be conserved, at least in higher plants. Prior to this work, only localization and/or retention mechanisms to the Golgi apparatus have been reported on enzymes related to protein glycosylation, glycan biosynthesis, SNAREs, and cytosolically oriented peripheral- and several integral-membrane proteins [34,35,36,37,38,39]. In some cases, glycan biosynthesis-related enzymes make protein complexes that allow the efficient reaction of glycan synthesis as well as the retention of the enzymes to the Golgi [35,39,40,41]. As type 2 P4H is a luminal protein that associates with the membrane tightly (Figure 2), it will be interesting to investigate whether type 2 P4Hs analyzed here make a protein complex that is related to glycan biosynthesis or other post-translational protein modifications in the Golgi.

Only a few proteins that do not have transmembrane regions were found in the Golgi lumen. One of them is the Cab45 protein found in mouse 3T3-L1 cells [42]. In this case, the mechanism of Golgi localization has not been elucidated. The other example is the precursor to type-I pectin methyl esterase in plants, which contains an N-terminal pro-region between the signal peptide and the mature protein [43]. A pro-region with multiple basic residues was shown to prevent the secretion of this protein from the Golgi, and proteolytic processing between the esterase domain and the pro-region directs the secretion of the enzyme into the extracellular space [43]. The third example is the case of GAUT1, a subunit of pectin synthetic complex [40]. Interestingly, the Tox1 domain and catalytic domain of type 2 P4H have no similarity with the pro-region of the pectin methyl esterase and with GAUT1. Therefore, the mechanism to retain P4Hs in the Golgi might be distinct from that of these proteins. Future analysis of the receptor to type 2 P4H in the Golgi lumen will be interesting in understanding the mechanism of protein retention in the Golgi.

The Tox1 domain of type 2 P4H is named based on the homology to BgK and ShK toxins, which are characterized as potassium channel blockers through the direct binding to channel proteins [12,13]. The Tox1 domain is also found in other organisms. It was reported that human microfibril-associated glycoprotein 1 and *Podocoryne carnea* astacin-like metalloproteinase contain C-terminal Tox1 domains [44]. Both these proteins contain N-terminal signal peptides, so they pass through the secretory pathway. In the case of microfibril-associated glycoprotein 1, this domain functions as a binding site for the extracellular matrix [45]. A similar domain, known as the ICR domain, is found in a number of cysteine-rich secretory proteins in mammals, and one of them from matrix metalloprotease 23 was shown to bind with a potassium channel to regulate its activity [46]. In all these cases analyzed, all the cysteine residues in the toxin homology domains are conserved and play a role in stabilizing the domain by forming disulfide bridges. These observations are consistent with the result that mutations in the cysteine residues caused the mis-localization of GFP-Tox1 proteins to the ER (Figure 8).

We observed that a type 2 P4H is associated with the membrane (Figure 1C,E) with the help of the Tox1 domain. Similarly, a GFP-Tox1 domain fusion, but not GFP alone, was tightly associated with the membrane (Figure 6B,C). Therefore, Tox1 domains, in general, have the ability to bind partner proteins. Interestingly, GFP fused with Tox1-truncated mutants of type 2 P4Hs, namely, GFP-NtP4H2.1 (ΔTox1) and GFP-NtP4H2.2 (ΔTox1), also seemed to localize in the Golgi (Figure 4H,I and Appendix A). These findings indicate that type 2 P4Hs are likely to have multiple signals for Golgi localization.

Another interesting observation is that both endogenous NtP4H2.2 and GFP-Tox1(NtP4H2.2) were not solubilized by Triton X-100 (Figure 1E and Figure 6F). This observation raises a possibility that the Tox1 domain has the ability to bind the microdomain of the membrane that is insoluble with Triton X-100 under low-temperature conditions [47]. However, we observed that Sec61, the major component of translocon in the ER, was not solubilized efficiently either (Figure 1E and Figure 6F). Therefore, further experiments, such as to solubilize membranes with other detergents as well as to test the solubility of other membrane protein(s), including Sec22, by these detergents will be necessary to determine the possibility that the Tox1 domain has the ability to associate the microdomain.

In this work, experiments showing Golgi localization of GFP-tagged type 2 P4H or Tox1 domains were performed by co-expression with RFP-tagged type 1 P4H, NtP4H1.1-mRFP. Our analysis of intracellular localization mechanisms of several Arabidopsis P4Hs indicated that some of the isoforms of P4Hs can localize in the ER when expressed alone, but these can be targeted to the Golgi by the interaction of other Golgi-localizing P4Hs [15]. Therefore, we cannot completely exclude the possibility that the co-expression of NtP4H1.1-mRFP reflects the distribution of these GFP-tagged proteins. However, we think this seems unlikely since GFP-tagged type 2 P4H and Tox1 domains showed a similar distribution when co-expressed with the non-Golgi protein SCAMP2-mRFP (Figure 4A–D, green). In the future, Golgi localization could be concluded by staining the expressing cells with antibodies against endogenous Golgi proteins, such as anti-NtP4H2.2, or by identifying receptors for type 2 P4H and/or Tox1 domains in the Golgi.

We observed that both type 1 [8] and type 2 (this work) P4Hs are predominantly localized in the Golgi. Both types of P4Hs are encoded by multigenes in higher plants. The primary sequences of the catalytic domains of these two classes of P4Hs are relatively distinct (Appendix A). Our previous analysis indicated that the catalytic domain of NtP4H1.1 has a weak role in retaining the protein in the ER [24], whereas the catalytic domain of NtP4H2.2 seemed to direct GFP to the Golgi (Figure 8). Thus, ancestral plants may have developed two different P4Hs in the secretory pathway, and both use different mechanisms in ER–Golgi localization. This raises the question of why plants developed these two genes and what is the difference between these two classes of P4H. A possibility is that the substrates for these enzymes are distinct. This idea is based on observations that recombinant type 1 P4H from Arabidopsis showed higher affinity for the poly(L-proline) than type 2 P4H from the same species [7]; recombinant Arabidopsis P4Hs showed different substrate specificity against non-contiguous proline residue in synthetic peptide [48]; and the prediction of substrate specificity suggests the difference in substrate specificity in three Arabidopsis P4Hs [15]. Future analyses of substrates for each isoform of P4H in vivo will reveal this possibility.

## 5. Conclusions

We analyzed the Golgi localization mechanism of plant type 2 prolyl hydroxylase, which is a peripheral membrane protein localized to the lumenal side of the Golgi and consists of catalytic and cysteine-rich Tox1 domains. We found that both the Tox1 domain and the catalytic domain contribute to the Golgi localization.

## Figures and Tables

**Figure 1 cells-13-01170-f001:**
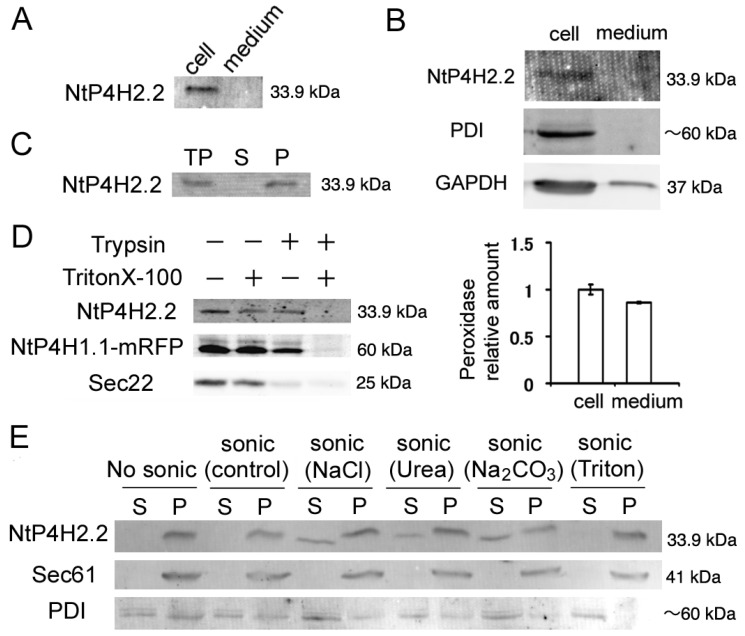
Membrane topology of NtP4H2.2. (**A**) The distribution of NtP4H2.2 in the cell and medium. Cellular proteins and medium proteins isolated from BY-2 culture were separated by SDS-PAGE, and the distribution of NtP4H2.2 was analyzed by immunoblotting using a NtP4H2.2-specific antibody. Both lanes correspond to the same amount of culture. (**B**) Distribution of NtP4H2.2 in tobacco protoplasts. Protoplasts prepared from WT BY-2 cells were grown for 3 days, and then a cellular fraction and a medium fraction were collected. Upper: Immunoblotting analysis of NtP4H2.2 and other cellular proteins (PDI and GAPDH). The same volume of total cellular protein fraction and medium fraction were electrophoresed and analyzed by immunoblotting using its specific antibodies. Lower: Peroxidase relative amount in cellular and medium fractions. The amount of peroxidase in the cellular fraction was set to 1. Data are means ± SD. (*n* = 3). (**C**) NtP4H2.2 was recovered in the precipitable fraction. Total proteins (TPs) isolated from BY-2 cells were separated into soluble (S) and precipitated (P) fractions by ultracentrifugation, and NtP4H2.2 and PDI were detected by immunoblotting. (**D**) Determination of the topology of NtP4H2.2. Microsomes prepared from BY-2 cells expressing NtP4H1.1-mRFP were treated with trypsin in the presence or absence of Triton X-100, and NtP4H2.2 and Sec22 were analyzed by immunoblotting or recording the fluorescence (NtP4H1.1-mRFP). (**E**) Membrane association of NtP4H2.2. Microsomes prepared from BY-2 cells were sonicated in buffers containing the chemicals as indicated, and then the soluble (S) and precipitated (P) fractions were separated by ultracentrifugation. Thereafter, the distribution of NtP4H2.2 and marker proteins (Sec61 and PDI) were analyzed by immunoblotting. PDI, protein disulfide isomerase, an ER soluble protein; Sec22, R-SNARE protein cycling between ER and Golgi apparatus; Sec61, an integral ER membrane protein. The migration position of each protein on SDS–polyacrylamide gel is indicated at the right of each panel.

**Figure 2 cells-13-01170-f002:**
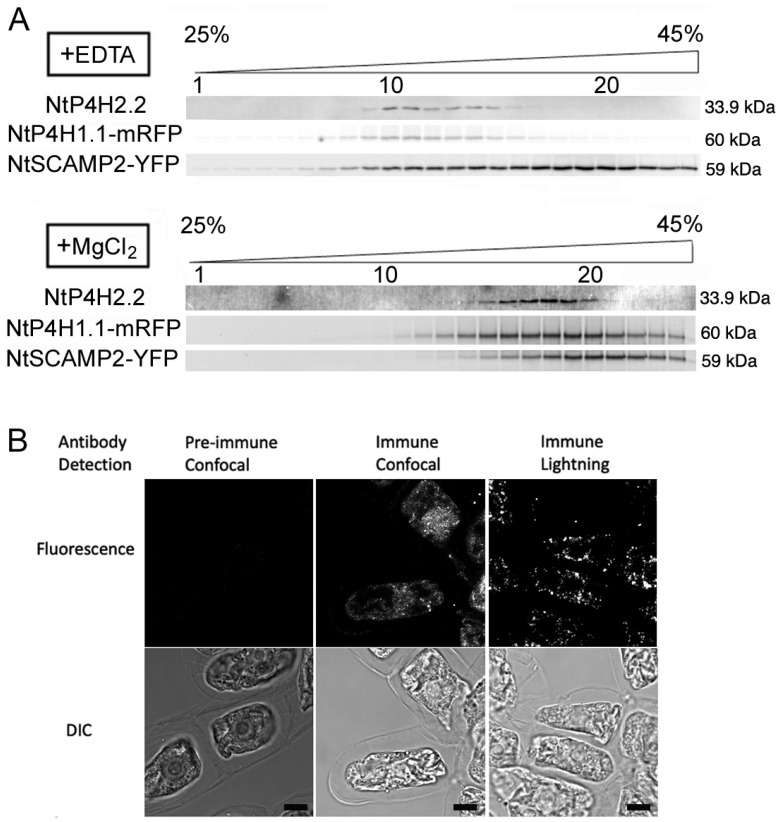
Intracellular localization analyses of endogenous NtP4H2.2. (**A**) Fractionation of endomembrane organelle. Microsomes were prepared from BY-2 cells expressing NtP4H1.1-mRFP and NtSCAMP2-YFP in the presence of EDTA (**A**) or MgCl2 (**B**) separated by isopycnic sucrose density gradient centrifugations and fractionated. After the separation of proteins, mRFP or YFP fluorescence was detected by an image analyzer. NtP4H2.2 was detected by immunoblotting using an NtP4H2.2-specific antibody. The numbers of fractions 1, 10, and 20 are indicated above the gel image. The concentration of sucrose in the gradient is shown at the top of the figure. The migration positions of each protein on SDS–polyacrylamide gel are indicated at the right of each panel. (**B**) Immunostaining of tobacco BY-2 cells with anti-NtP4H2.2 antibody. Fixed tobacco BY-2 cells were incubated with either pre-immune serum or anti-NtP4H2.2 antibody with fluorescent-tagged secondary antibody. Confocal and lightning super-resolution images are shown. Bars in B = 10 µm.

**Figure 3 cells-13-01170-f003:**
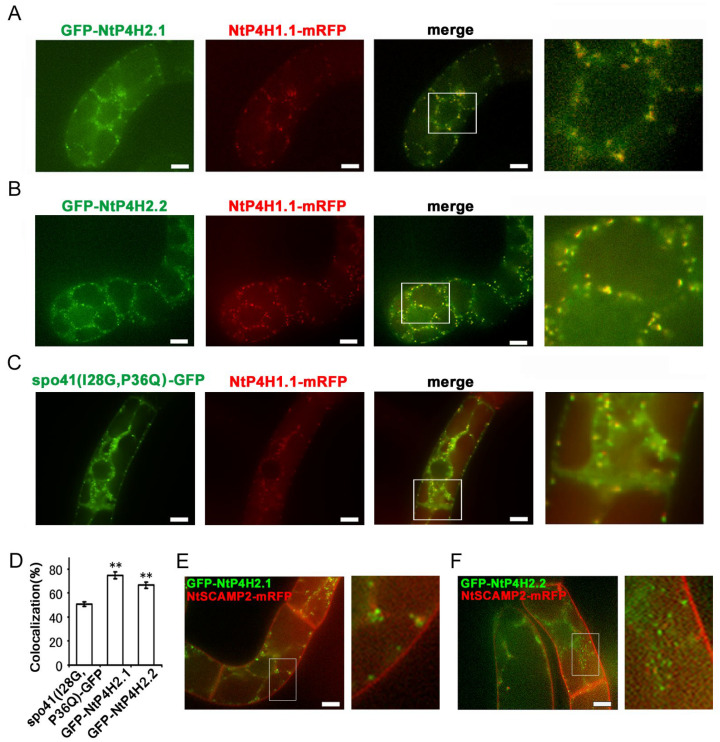
Localization of GFP-type 2 P4H fusion proteins in tobacco BY-2 cells. GFP-fusion proteins were expressed in BY-2 cells expressing NtP4H1.1-mRFP. Confocal overlay images of GFP-NtP4H2.1 (**A**), GFP-NtP4H2.2 (**B**), or spo41(I28G,P36Q)-GFP (**C**) with NtP4H1.1-mRFP are shown. The green fluorescence (far left) and red fluorescence (second from the left) images are shown, along with a merged image of these two colors (third from the left). An enlarged image of the boxed area in the merged image (far right) is also shown. (**D**) Colocalization frequencies of GFP and RFP signals. GFP and RFP fluorescent intensities in a randomly chosen fluorescent dot were obtained by a line scan program in Metamorph software. Colocalization frequencies of GFP and RFP signals in randomly chosen dots were calculated. The data are the means ± SD (*n* = 18). Double asterisks indicate a significant difference between spo41(I28G,P36Q)-GFP and type 2 P4H fusion proteins (*p* < 0.01). GFP-fusion proteins were expressed in BY-2 cells expressing NtSCAMP2-mRFP. Confocal overlay images of GFP-NtP4H2.1 (**E**) or GFP-NtP4H2.2 (**F**) with NtSCAMP2-mRFP and enlarged images of the boxed areas of merged images are shown to the right of each image. Bars in (**A**–**C**,**E**,**F**) = 10 µm.

**Figure 4 cells-13-01170-f004:**
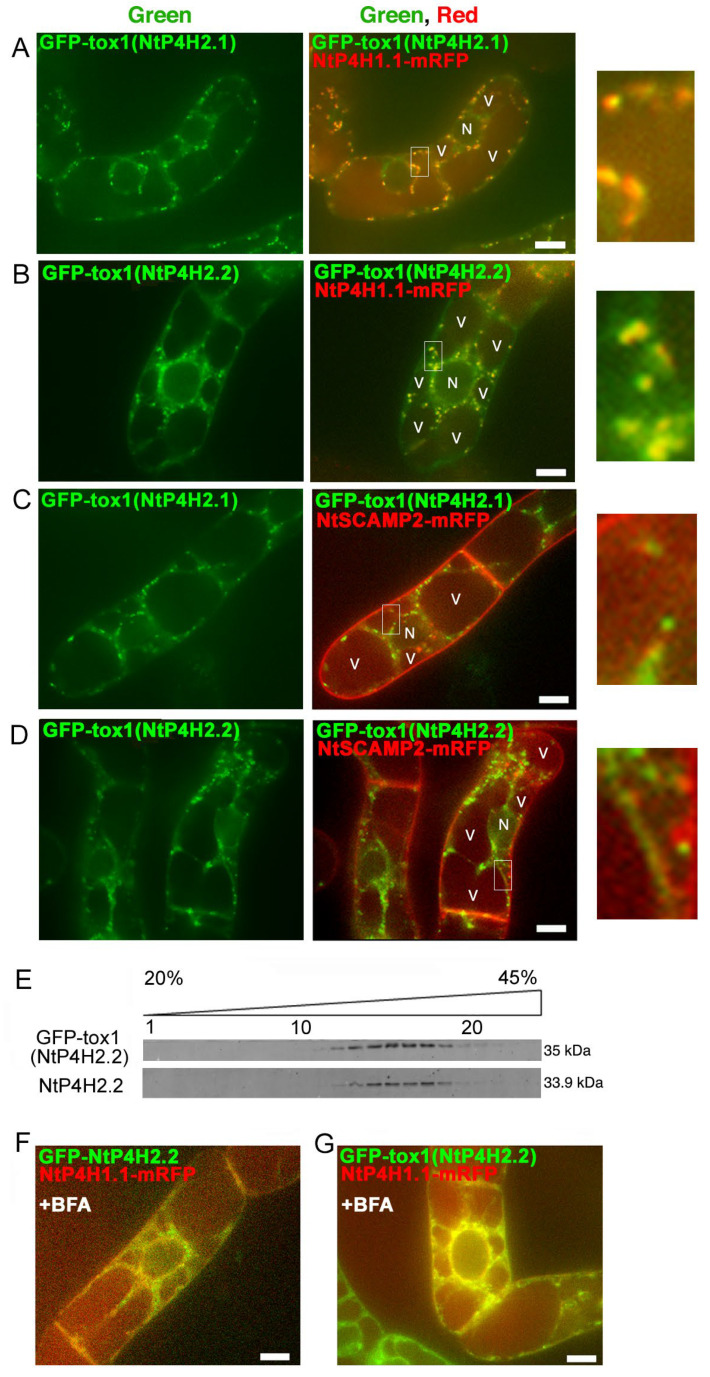
Localization of GFP-Tox1 and GFP-catalytic domain fusions in BY-2 cells. GFP-Tox1 fusion proteins were expressed in BY-2 cells expressing NtP4H1.1-mRFP. Confocal green fluorescence (left) and overlay images of GFP-Tox1(NtP4H2.1) (**A**) or GFP-Tox1(NtP4H2.2) (**B**) with NtP4H1.1-mRFP (center), as well as enlarged images of the boxed areas of merged images (right) are shown. GFP-Tox1 fusion proteins were expressed in BY-2 cells expressing NtSCAMP2-mRFP. Confocal green fluorescence (left), overlay images of GFP-Tox1(NtP4H2.1) (**C**) or GFP-Tox1(NtP4H2.2) (**D**) with NtSCAMP2-mRFP (center), and enlarged images of the boxed areas of merged images (right) are shown. (**E**) Membrane fractionation analysis in the presence of EDTA of BY-2 cells expressing GFP-Tox1(NtP4H2.2). GFP-Tox1 and NtP4H2.2 signals were detected by immunoblotting using specific antibodies. The numbers above the blot images are the fraction numbers. The concentration of sucrose in the gradient is shown at the top of the image. Effects of brefeldin A (BFA) on GFP-NtP4H2.2 (**F**) and GFP-Tox1(NtP4H2.2) (**G**) localization in BY-2 cells expressing NtP4H1.1-mRFP. Cells were treated with 14.8 µM BFA for 120 min. After the BFA treatment, their distributions changed from the Golgi pattern to the ER pattern. Bar = 10 µm. N and V in (**A**–**D**) indicate the nucleus and vacuole, respectively.

**Figure 5 cells-13-01170-f005:**
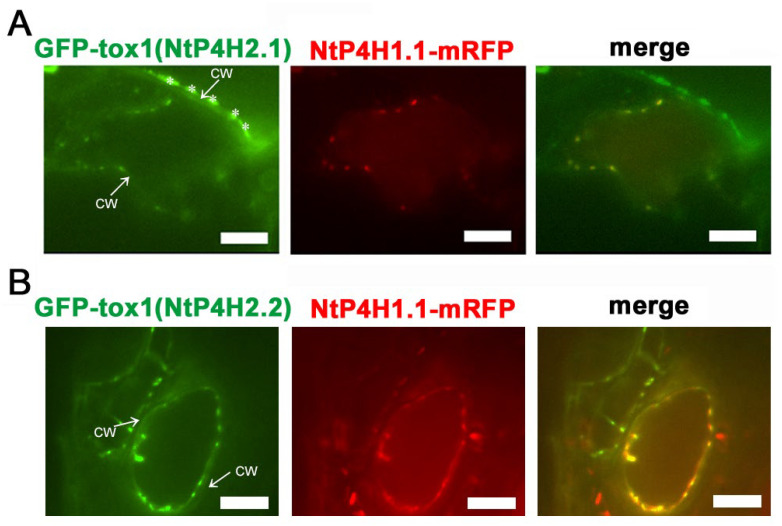
Localization of Tox1-fused GFP in Arabidopsis cells. Either GFP-Tox1(NtP4H2.1) (**A**) or GFP-Tox1(NtP4H2.2) (**B**) and NtP4H1.1-mRFP were transiently expressed in Arabidopsis leaf epidermal cells. Confocal individual color images and merged images are shown. The arrow with CW indicates the position of the cell wall. Asterisks indicate auto-fluorescent structures observed outside the cell. Scale bar = 10 µm.

**Figure 6 cells-13-01170-f006:**
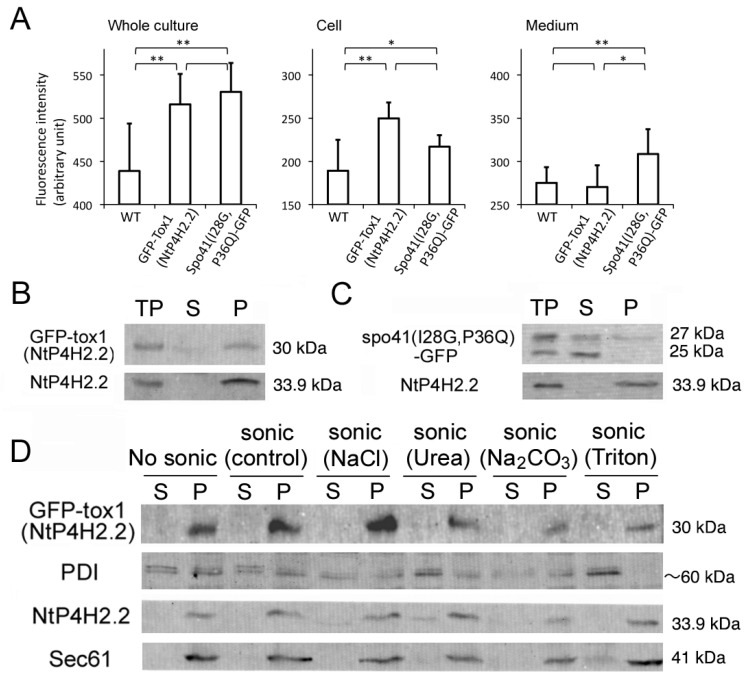
The Tox1 domain increases the membrane association of GFP. (**A**) Cell and medium distribution of GFP-Tox1 fusion and control. Fluorescence of whole culture, cell, and medium fractions of non-transformed control cells (WT) expressing GFP-Tox1(NtP4H2.2) or spo41(I28G,P36Q)-GFP were measured at 488 nm excitation and 520 nm emission. An average of six independent samples is shown. The error bar indicates the standard deviation. Single and double asterisks indicate significant differences at (*p* < 0.05) and (*p* < 0.01), respectively. (**B**,**C**) The distributions of spo41(I28G,P36Q)-GFP and GFP-Tox1(NtP4H2.2) in the cell. Tobacco cells expressing either GFP-Tox1(NtP4H2.2) (**B**) or spo41(I28G, P36Q)-GFP (**C**) were fractionated as described in the Figure 1B legend, and GFP and control proteins (NtP4H2.2) were detected by immunoblotting using specific antibodies (TP, total protein; S, soluble fraction; P, precipitated fraction). (**D**) Membrane association of GFP-Tox1. Microsomes isolated from BY-2 cells expressing GFP-Tox1(NtP4H2.2) were sonicated in buffers containing the indicated compounds, then soluble (S) and precipitated (P) fractions were separated by ultracentrifugation. GFP and control proteins (PDI, NtP4H2.2, and Sec61) were detected by immunoblotting using specific antibodies. The migration position of each protein on SDS–polyacrylamide gel is indicated to the right of each panel.

**Figure 7 cells-13-01170-f007:**
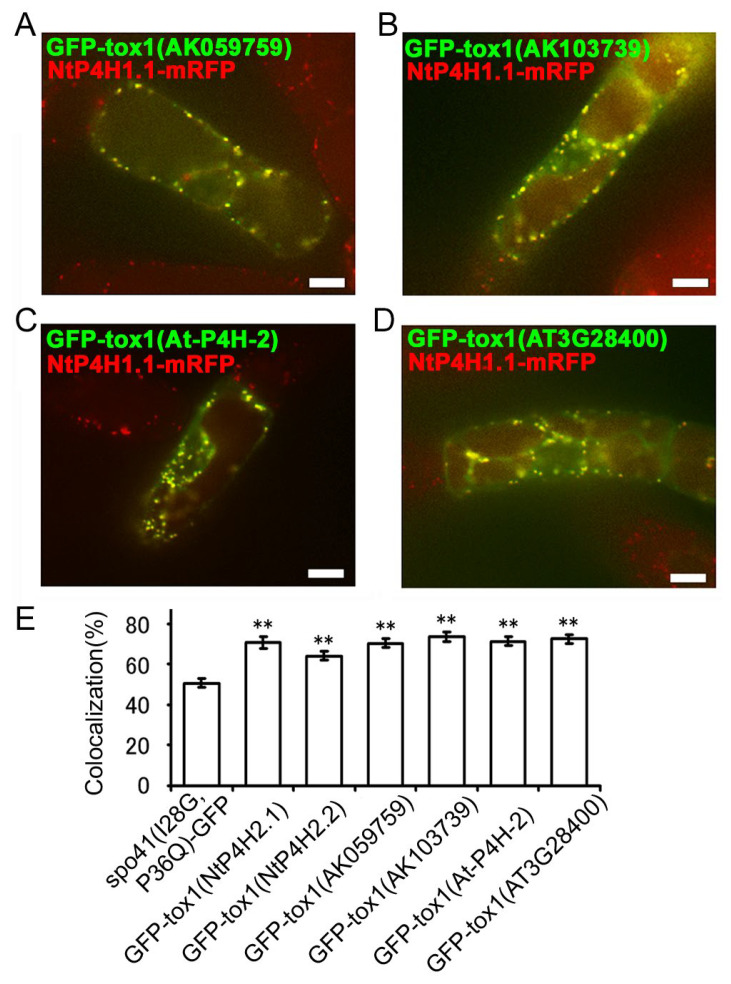
Localizations of GFP fused with the Tox1 domain from type 2 P4H homologs of various plants. The Tox1 domains from rice (AK059759 and AK103739) or Arabidopsis (At-P4H-2 and AT3G28480) were fused with spo41(I28G,P36Q)-GFP and expressed in BY-2 cells expressing NtP4H1.1-mRFP. Confocal overlay images of GFP-Tox1 (AK059759) (**A**), GFP-Tox1 (AK103739) (**B**), GFP-Tox1 (At-P4H-2) (**C**), or GFP-Tox1 (AT3G28480) (**D**) with NtP4H1.1-mRFP are shown. Bar = 10 µm. (**E**) Colocalization frequencies of GFP and RFP signals. Data were collected as described in the legend in Figure 3D. Double asterisks indicate a significant difference (*p* < 0.01). (**E**) Colocalization frequencies of GFP-Tox1 and RFP signals, as described in the Figure 3D legend. Data are the means ± SD (*n* = 18). Double asterisks indicate a significant difference between spo41(I28G,P36Q)-GFP and GFP-Tox1 fusion proteins (*p* < 0.01).

**Figure 8 cells-13-01170-f008:**
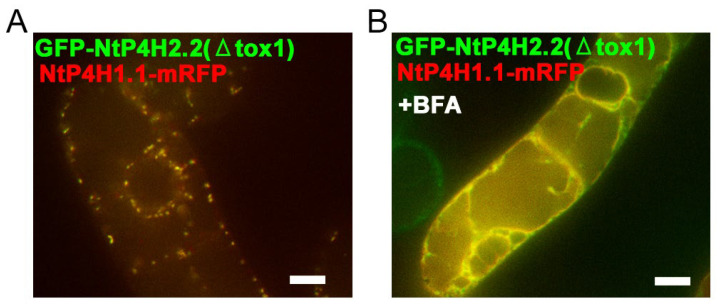
Localization of GFP-NtP4H2.2 catalytic domain fusion in BY-2 cells. Tox1-truncated GFP-NtP4H2.2 was expressed in BY-2 cells expressing NtP4H1.1-mRFP. (**A**) Confocal overlay images of GFP-GFP-NtP4H2.2(ΔTox1) with NtP4H1.1-mRFP are shown. (**B**) Effects of BFA on GFP-NtP4H2.2(ΔTox1) localization in BY-2 cells expressing NtP4H1.1-mRFP. Cells were treated with 14.8 µM BFA for 120 min. After the BFA treatment, their distributions changed from the Golgi pattern to the ER pattern. Bar = 10 µm.

**Figure 9 cells-13-01170-f009:**
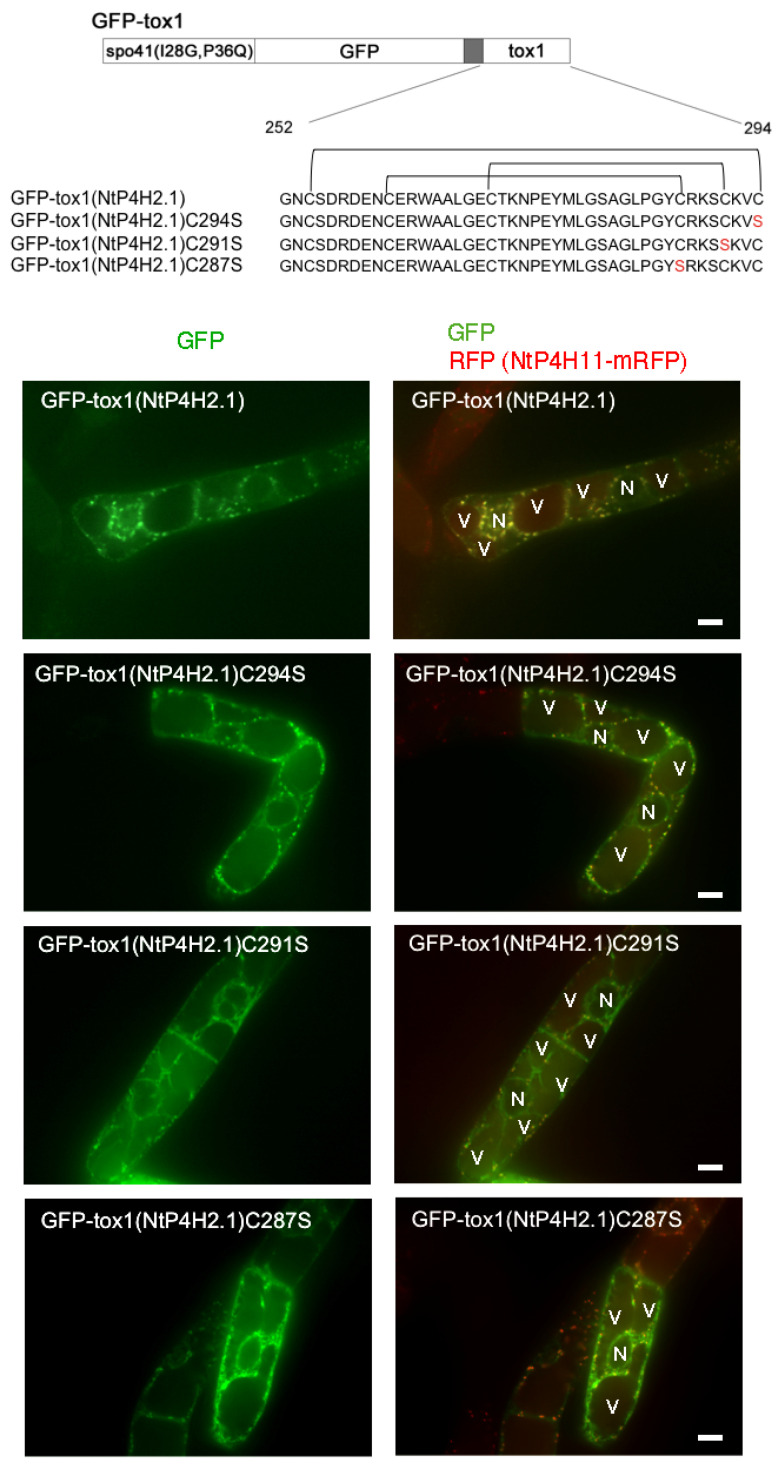
Mutations of cysteine residues in the Tox1 domain caused the accumulation of GFP-Tox1 protein in the ER. Structures of GFP fusion constructs with wild-type and mutant Tox1 domains are shown at the top. Predicted disulfide bridges are indicated above the sequence. These proteins were transiently expressed in BY-2 cells expressing NtP4H1.1-mRFP, and fluorescence in the transiently transformed cells was recorded. Confocal images of GFP fluorescence (left) and merged fluorescence of GFP and RFP (right) of representative cells are shown. N, nucleus; V, vacuole. Bar = 10 µm.

## Data Availability

The original contributions presented in the study are included in the article/Appendix A, further inquiries can be directed to the corresponding author.

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
