# Peer review of "Toxin Homology Domain in Plant Type 2 Prolyl 4-Hydroxylases Acts as a Golgi Localization Domain"

_cells, 2024, doi:10.3390/cells13141170_

Round 1
Reviewer 1 Report
Comments and Suggestions for Authors
This manuscript reports the discovery of two novel protein domains responsible for localizing soluble proteins in the lumen of the plant Golgi apparatus. This significant finding warrants publication.
The authors demonstrated the localization of two PH4 proteins, NtP4H2.2 and NtP4H2.1, at the Golgi by comparing their localization with the previously characterized Golgi-localized PH4, NtP4H1.1-mRFP, through co-transfection experiments with GFP-tagged constructs.
Furthermore, they showed that NtP4H2.2 is localized in the luminal region of microsomes isolated from BY-2 cells, as evidenced by trypsin digestion only in the presence of Triton-X100.
To investigate the components of NtP4H2.2 necessary for Golgi localization, the authors fused GFP to the Tox1 domain of NtP4H2.2, which is also present in other PH4 proteins. They found that Tox1 alone is sufficient to direct GFP to the Golgi when co-transfected with NtP4H1.1-mRFP, observed in both BY-2 and A. thaliana cells. The Golgi-targeting ability of Tox1 appears to be conserved across species, as indicated by GFP-P4H constructs from other plant species. Additionally, Tox1 was shown to follow Golgi sub-compartments (to the ER and PM) upon BFA treatment.
The authors also observed Golgi localization when GFP was fused to the catalytic domain of NtP4H2.2 (without the Tox1 domain), suggesting the involvement of the catalytic domain in enzyme localization.
To delve further into the characteristics of the NtP4H2.2 Tox1 domain facilitating Golgi localization, the authors investigated the role of conserved cysteines through C-to-S mutations. These mutations resulted in a change in GFP-NtP4H2.1 localization from Golgi to a more tubular ER pattern.
Overall, this paper offers valuable insights into the Golgi-localizing function of the Tox1 and catalytic domains of NtP4H2.2. While the manuscript is well-written and controlled, the inclusion of additional controls and the use of endogenous markers would enhance its value.
Here are some suggestions for improving the clarity and completeness of the manuscript:
1. Figures 1 and 6 both assess the localization of hybrid protein constructs in the Golgi lumen using various treatments. While the data is generally clear, there are issues with Triton X-100 treatment not solubilizing the transmembrane Sec61 protein effectively. Including a control with a transmembrane Golgi protein like Sec22 would provide a helpful comparison.
2. In Figure 1, the authors state that NtP4H2.2 is not secreted. However, it's unclear how the cell and medium protein concentrations were normalized, which raises questions about whether levels of NtP4H2.2 might be higher in the medium but more diluted than in the cellular fraction.
3. Figure 2 is missing the 2C designation.
4. No endogenous Golgi marker was utilized throughout the paper to confirm Golgi localization. Using anti-NtP4H2.2 antibodies for localization of endogenous NtP4H2.2 (or any other endogenous Golgi marker) would provide more informative data than relying solely on transient expression of tagged-NtP4H1.1, which is claimed to be Golgi-localized.
5. The authors assert that both Tox1 and the catalytic domains are sufficient for Golgi localization of NtP4H2.2. However, only one data point is provided to support this claim for the catalytic domain (Figure 8).
6. In the datasets for BFA treatment, the authors claim that their NtP4H2.2 construct follows the Golgi to the ER. However, this assertion lacks supporting evidence from an endogenous ER marker.
7. For constructs with Tox1, as no Golgi marker was used, the authors could have fused a tag to known sequences that localize at the Golgi as a control.
Reviewer 2 Report
Comments and Suggestions for Authors
This manuscript provides new insights into the function of a toxin homology domain in the subcellular localisation of plant prolyl 4-hydroxylases (P4Hs). An antibody specific for an endogenous P4H was generated and used to analyse the subcellular distribution and GFP-tagged versions of P4H or only the TOX1 domain were expressed in different plant species. Overall, the data and conclusions presented are very clear, the results are of great interest as these TOX domain proteins are highly conserved and I have only a few minor issues that need to be addressed:
Title and also in the text: do the authors have sufficient evidence to conclude that the TOX domain acts as a Golgi targeting domain and not as a Golgi retention domain? The otherwise secreted spoGFP is retained in the Golgi when the TOX domain is added. The cysteine mutants appear to be located in the ER, which could be a consequence of misfolding and not because a potential targeting motif in the TOX domain has been mutated.
In many Figures only the co-localisation is shown. Please add also for some expressed constructs like GFP-tox1 images without any co-expressed marker. In addition, please perform co-localisaiton of GFP-tox1 with a more unrelated RFP-Golgi marker to rule out that NtP4H1.1 expression has an impact on GFP-tox1.
Figure with immunoblot data: please indicate the size of the proteins (in kDa).
Line 276: What does “associated with organelles” mean? This was not really investigated. Please change the wording.
Line 324: microsomes instead of microsome
Line 457: add space between nm and excitation
Lines 490 and 491: E-value instead of e-value
Lines 513 and 515: AT3G28480 instead of AT3G028480
Line 576: the domain could be important for protein folding which subsequently affects the function.
Line 579: can you conclude that they indeed cycle between the ER and Golgi? BFA results in redistribution to the ER but this is more an unspecific event for Golgi proteins and does not give hints on the cycling.
Line 600: remove “of”
Line 602: in some cases
Line 604: N-glycan processing enzymes form also complexes in the Golgi (see for example https://doi.org/10.1104/pp.112.210757)
Line 605: that associates
Line 608: that do not
Lines 617-619: you may also mention GAUT1 and its retention mechanism as another example (https://doi.org/10.1073/pnas.1112816108)
Line 635: may be better signals instead of sites?
Line 642: by multigenes
Figure 3C: Why is spoGFP detected in the Golgi? It should be efficiently secreted. Please provide an explanation.
Figure 3E and F: add an inset to better show the dots appearing in close proximity.
Figure 4F and G: the images are brighter this appears a bit strange, can you reduce the brightness of the images?
Figure 5B: are this leaf epidermal cells? Please specify. Why are there only few Golgis? Do you have other images?
Figure 9: the network-like structure is not very well visible. Could you provide other images or do a co-localisation with an ER-marker?
Figure 9: do the GFP-tox1 mutants have the same expression levels like GFP-tox1? Can you show an immunoblot with anti-GFP antibodies?
Round 2
Reviewer 1 Report
Comments and Suggestions for Authors
I understand the author’s difficulties in performing additional experiments indicated in my review. As I’ve mentioned in my review, these experiments would improve the paper but will not change the major conclusions, which are very significant for trafficking/organelle/protein localization. Therefore, I would like to confirm my second review decision is "Accept".
Reviewer 2 Report
Comments and Suggestions for Authors
The authors have addressed most of my concerns and I have no further suggestions for improvement.